# Vitamin D Status and Psoriatic Arthritis: Association with the Risk for Sacroiliitis and Influence on the Retention Rate of Methotrexate Monotherapy and First Biological Drug Survival—A Retrospective Study

**DOI:** 10.3390/ijms24065368

**Published:** 2023-03-10

**Authors:** Cinzia Rotondo, Francesco Paolo Cantatore, Daniela Cici, Francesca Erroi, Stefania Sciacca, Valeria Rella, Addolorata Corrado

**Affiliations:** Rheumatology Unit, Department of Medical and Surgical Sciences, University of Foggia, 71122 Foggia, Italy

**Keywords:** psoriatic arthritis, DMARDs, biological drugs, methotrexate, vitamin D deficiency or insufficiency

## Abstract

A growing body of evidence on the importance of vitamin D in immune modulation has increased the interest in its possible impact on the course of rheumatological diseases. The scope of our study is to assess if the presence of different statuses of vitamin D could interfere in the clinical subsets, in methotrexate monotherapy discontinuation, and biological drug (b-DMARDs) survival in psoriatic arthritis patients (PsA). We conducted a retrospective study on PsA patients and split them into three groups based on their vitamin D status: the group with 25(OH)D ≤ 20 ng/mL, the group with levels of 25(OH)D between 20 and 30 ng/mL, and the group with serum levels of 25(OH)D ≥ 30 ng/mL. All patients were required to fulfill the CASPAR criteria for psoriatic arthritis and to have the evaluation of vitamin D serum levels at baseline visit and at clinical follow-up visits. The exclusion criteria were ages less than 18 years old, the presence of HLA B27, and satisfaction of rheumatoid arthritis classification criteria (during the study time). Statistical significance was set at *p* ≤ 0.05. Furthermore, 570 patients with PsA were screened and 233 were recruited. A level of 25(OH)D ≤ 20 ng/mL was present in 39% of patients; levels of 25(OH)D between 20 and 30 ng/mL presented in 25% of patients; 65% of patients with sacroiliitis presented 25 (OH)D ≤ 20 ng/mL. Methotrexate monotherapy discontinuation for failure was higher in the group with 25 (OH)D ≤ 20 ng/mL (survival time: 92 ± 10.3 weeks vs. 141.9 ± 24.1 weeks vs. 160.1 ± 23.6 weeks; *p* = 0.02) with higher discontinuation risk (HR = 2.168, 95% CI 1.334, 3.522; *p* = 0.002) than those with 25(OH)D between 20 and 30 ng/mL and those with 25(OH)D ≥ 30 ng/mL. Significantly shorter survival of first b-DMARDs was assessed in the group with 25 (OH)D ≤ 20 ng/mL versus the other groups (133.6 ± 11 weeks vs. 204.8 ± 35.8 weeks vs. 298.9 ± 35.4; *p* = 0.028) (discontinuation risk 2.129, 95% CI 1.186, 3.821; *p* = 0.011). This study highlights significant differences in clinical presentation, in particular sacroiliac involvement and on drug survival (methotrexate and b-DMARDs) in PsA patients with vitamin D deficiency. Further prospective studies, including a larger sample of patients, are needed to validate these data and to assess if the supplementation of vitamin D could improve the b-DMARDs response in PsA patients.

## 1. Introduction

Psoriatic arthritis (PsA) is a chronic inflammatory disease characterized by wide heterogeneity of clinical domains (enthesitis, dactylitis, cutaneous involvement, possible presence of axial arthritis with inflammatory back pain, uveitis, and inflammatory bowel diseases) and a huge variety of clinical courses. The hallmarks of PsA are distal interphalangeal joint involvement and new bone apposition at X-rays. A growing body of evidence on clinical presentation with extra-cutaneous and extra-articular manifestation and a new hypothesis on possible autoimmune pathogenesis with the evidence of specific autoantibodies emphasized the theory of “Systemic Psoriatic Disease” [1,2,3,4,5,6].

Vitamin D, as well as playing a crucial role in the homeostasis of calcium–phosphorus metabolism and in bone health, seems to act on the proper function of the immune system, inhibiting the synthesis of macrophage-derived cytokines (Il-1, IL-6, IL-12, TNF-α), and suppressing the Th1 cells production of IL-2 [7,8,9,10]. Although these functions involve the proven role of vitamin D in infection prevention, including its function in the cardiovascular system through important renoprotection activity, as well as the prevention of cancer and autoimmune diseases such as diabetes mellitus, rheumatoid arthritis, and multiple sclerosis, which share Th1/Th2 dysregulation and other immunologic abnormalities with psoriasis [11,12,13], agreement on optimal levels of 25 (OH) D has not yet been reached [14,15,16].

A lot of scientific evidence has described a higher rate of vitamin D deficiency in psoriasis and spondyloarthritis patients [17,18,19,20,21]. Contrasting data are reported on the possible association between lower levels of vitamin D and inflammation status or disease activity in PsA patients [20,21,22,23,24]. Regardless, the possible role of vitamin D on the IL-12-IL-23 cytokines axis and the IL-17 pathway could partially explain how vitamin D deficiency acts on the PsA disease mechanism and spondylarthritis [10].

Furthermore, few data, all on inflammatory bowel diseases, are reported on the possible association between the insufficiency of vitamin D and biological disease-modifying anti-rheumatic drugs (b-DMARDs) response, with the majority of evidence in favor of poor response to anti-tumor necrosis factor-alpha (anti TNF-α) in patients with vitamin D insufficiency [25,26,27].

Based on these discrepancies, how vitamin D deficiency can interfere with the clinical course and drug response in PsA is not yet understood or has not been evaluated. This study aims to assess, in the real-life clinical setting, if various serum statuses of vitamin D can determine different PsA clinical subsets and can impact methotrexate monotherapy survival and first b-DMARDs retention rate.

## 2. Results

In total, 233 out of 570 PsA patients evaluated satisfied the inclusion criteria of the study (154 female and 79 male) (337 patients were excluded, 25% did not satisfy CASPAR criteria, 35% had no periodic vitamin D evaluations, 30% changes their vitamin D status in the follow-up period, 10% was positive for HLA B27). The mean age of patients was 57.3 *±* 13.4 years, with a symptom duration of 4.6 *±* 5.3 years at the PsA diagnosis time. Levels of 25(OH)D ≤ 20 ng/mL were present in 39% of patients (mean serum level 13.7 *±* 4.9 ng/mL), and levels of 25(OH)D between 20 and 30 ng/mL in 25% of patients (mean serum level 25.3 *±* 2.5 ng/mL). Regarding the persistent low serum level of 25(OH)D, despite the doctor’s suggestion of supplementation, all patients declared poor or no adherence to the suggested vitamin D supplementation therapy. No patient was taking active metabolites of vitamin D.

No patients with hypovitaminosis D presented osteomalacia based on clinical and serological characteristics (in particular parathyroid hormone sera levels and alkaline phosphatase levels) and the normal value of bone mineral density at densitometry.

The comparisons of clinical features between the groups of PsA patients in the study at baseline are reported in Table 1. No significative differences were found in the rate of methotrexate (MTX) monotherapy, the rate of articular subsets (oligo-articular, poly-articular), and enthesitis among the groups of patients in the study (Table 1). Of note, all patients presented a good mobilization degree.

The group of patients with 25 (OH)D ≤ 20 ng/mL serum level was characterized by a significantly higher rate (65%) of asymmetrical sacroiliitis (*p =* 0.0001) (Table 1). The estimated risk for sacroiliitis associated with 25 (OH)D ≤ 20 ng/mL serum level was high (OR 5.8; 95% CI 2.419, 14.290; *p =* 0.001).

No significant correlations were found between serum levels of vitamin D and acute phase reactants such as ERS (r *=* 0.83; *p =* 0.530) and CRP (r *=* −0.002; *p =* 0.989). A negative correlation was found between vitamin D serum level and the degree of sacroiliitis (r *=* −0.267; *p =* 0.0001).

As regards comorbidities, we found a higher trend of inflammatory bowel diseases (Crohn’s disease or ulcerative colitis), diabetes, osteoporosis, and cardiovascular diseases in the group with 25 (OH)D ≤ 20 ng/mL serum level compared to the group with 25(OH)D ≥ 30 ng/mL serum level (Table 1). Furthermore, we found a higher total cholesterol/HDL ratio in patients with 25 (OH)D ≤ 20 ng/mL serum level compared to those with 25(OH)D ≥ 30 ng/mL serum level (*p =* 0.040), in particular in the male group (Table 1).

### 2.1. Methotrexate Monotherapy Treatment Survival and Vitamin D Serum Levels in Psoriatic Arthritis Patients

One hundred and eighty-nine patients (81%) started monotherapy with MTX (15 mg/weekly) at the diagnosis time. None of the patients who started therapy with MTX had an oligo-articular subset or sacroiliitis.

In the whole group of patients, the estimation of MTX monotherapy survival time was 207.5 *±* 16.7 weeks. Of note, MTX failure was registered in 72% of patients with polyarticular involvement and 70% of patients with enthesitis subset.

There was a significant correlation between serum levels of vitamin D and MTX treatment duration (r *=* 0.341; *p =* 0.0001).

Considering different groups of PsA patients in the study, overall MTX monotherapy survival tended to be shorter in patients with 25 (OH)D ≤ 20 ng/mL serum level and those with 25 (OH)D between 20 and 30 ng/mL serum level than those with 25(OH)D ≥ 30 ng/mL serum level (119.8 *±* 9.1 weeks vs. 163.4 *±* 21.1 weeks vs. 239.2 *±* 27.7 weeks, respectively; *p =* 0.098) (Table 2).

Evaluating the MTX discontinuation for the failure of monotherapy, we evidenced that MTX monotherapy survival was significantly shorter in the group with 25 (OH)D ≤ 20 ng/mL serum level (92 *±* 10.3 weeks vs. 141.9 *±* 24.1 weeks vs. 160.1 *±* 23.6 weeks; *p =* 0.02) with higher discontinuation risk (HR *=* 2.168, 95% CI 1.334, 3.522; *p =* 0.002) compared to the group with 25 (OH)D between 20 and 30 ng/mL serum level and the group with 25(OH)D ≥ 30 ng/mL serum level, respectively (Figure 1A), although no differences in the rate of MTX discontinuation for failure were found among the groups in the study (*p =* 0.340) (Table 2). All MTX-failure patients required combination therapy with b-DMARDs.

Considering the patients that discontinued MTX for adverse events, no significative differences in MTX monotherapy survival time were observed among different groups in the study, 67.7 *±* 15.4 weeks, 83.2 *±* 17.6 weeks vs. 136.6 *±* 41.3; *p =* 0.661, respectively, in the patients with 25 (OH)D ≤ 20 ng/mL, those with 25 (OH)D between 20 and 30 ng/mL and those with 25(OH)D ≥ 30 ng/mL serum level (Figure 1B).

The gender analysis, based on stratification in males and females, showed no changes in the retention rate of the first b-DMARDs treatment with respect to the result obtained in the whole group (*p =* 0.115).

### 2.2. First b-DMARDs Treatment Retention Rate and Vitamin D Serum Levels in Psoriatic Arthritis Patients

One hundred thirty-four patients started b-DMARDs during follow-up, 88% after MTX failure, and 12% as the first treatment; the specifications are displayed in Table 2.

A significant correlation between serum levels of vitamin D and b-DMARDs treatment duration (r *=* 0.255; *p =* 0.007).

A shorter first b-DMARDs survival time was found in the patients with 25 (OH)D ≤ 20 ng/mL serum level compared to those with 25 (OH)D between 20 and 30 ng/mL and those with 25(OH)D ≥ 30 ng/mL serum level (133.6 *±* 11 weeks vs. 204.8 *±* 35.8 weeks vs. 298.9 *±* 35.4; *p =* 0.028) (Figure 2). Using Cox regression analysis, the discontinuation risk of first b-DMARDs was higher in the group with 25 (OH)D ≤ 20 ng/mL serum level (HR *=* 2.129, 95% CI 1.186, 3.821; *p =* 0.011) compared to the other study groups. The failure of the first b-DMARDs was revealed in 18 patients with 25 (OH)D ≤ 20 ng/mL serum level, in 17 patients with 25 (OH)D between 20 and 30 ng/mL serum level, and in 32 patients with 25(OH)D ≥ 30 ng/mL serum level. Most of the patients (95%) that discontinued the first b-DMARDs were treated with anti TNF-α; only 4% of these patients were treated with anti-IL-17 drugs (Table 2).

The gender analysis, based on stratification in males and females, showed no changes in the retention rate of the first b-DMARDs treatment with respect to the result obtained in the whole group (*p =* 0.153).

## 3. Discussion

The established protective role of vitamin D in cancer and immune disorder [8], the new hypothesis on autoimmune pathogenesis of PsA, and the proof of specific psoriatic autoantibodies, which consistently support the theory of “Systemic Psoriatic Disease” [1,2,3,4], greatly increase the interest on the possible function of vitamin D in PsA.

A higher prevalence of vitamin D insufficiency is commonly described in psoriatic patients with or without arthritis, with some correlations between vitamin D deficit and disease activity, markers of inflammation, and comorbidities [17,18,19,20,21,28,29,30,31], although some contrasting data are reported [20,24].

We have realized a retrospective study to assess the possible relations between the different statuses of vitamin D and clinical features of psoriatic arthritis (PsA) and DMARDs drug survival.

Overall, our results partially echo previous studies. We evidenced no significant differences in PsA disease activity, evaluated using clinimetric scores as DAS and DAPSA at baseline [20,24]. However, we noted a high percentage of patients with sacroiliitis in the group with 25 (OH) D ≤ 20 ng/mL, finding a significant association between low serum level of 25 (OH) D and sacroiliitis in PsA patients with a higher odds ratio (OR 5.8; 95% CI 2.419, 14.290; *p =* 0.001). This is in line with previous studies on the correlation between disease severity and the status of vitamin D, in which the presence of sacroiliitis is considered as more severity of the disease [32]. No data have previously described the risk of sacroiliitis and the deficit of vitamin D in PsA patients. Some studies have reported an association between sacroiliitis and osteomalacia, but none of our patients had characteristics of osteomalacia. A published study on spondylarthritis evidenced a higher percentage of vitamin D insufficiency in patients with spondylarthritis and sacroiliitis, hypothesizing that although the structural damage due to sacroiliitis usually does not determine mobility limitations, the highest frequency of spine involvement in patients with spondylarthritis and sacroiliitis might cause mobility limitation and inability with potentially decreased sun exposure [32]. None of our patients presented spine involvement or functional limitations with inability; thus, we cannot support these hypotheses. The physiopathological mechanism is not yet clarified. Previous authors have hypothesized that the sacroiliitis could be due to subchondral bone abnormalities associated with metabolic bone diseases [23]. Furthermore, a possible action of vitamin D is more conceivable, with probable influence on immunological response and different cytokines axes and interleukins pathways (as IL-12/IL-23 and Il-17) involved in bone damage in PsA. Furthermore, the potential action of vitamin D on the WNT/β-catenin signaling pathway [33,34] also in PsA patients could be taken into account to explain this relationship, but further in vivo and in vitro studies need to be performed to understand the real mechanisms of the association between sacroiliitis and vitamin D status.

The crucial novelty of this study is the association between the lowest serum level of 25(OH)D and the risk of DMARDs discontinuation.

To the best of our knowledge, this is the first study on b-DMARDs drug survival in PsA patients and deficiency of vitamin D. Published studies on poor response to anti-TNF α treatment in the presence of insufficiency of vitamin D have only involved inflammatory bowel disease (IBD) patients. In particular, in a study conducted on 76 patients with IBD, the authors evidenced that in Crohn’s disease patients, the presence of 25 (OH) D < 30 ng/mL, in particular during the induction phase, was significantly related to early termination of anti-TNF therapy (14.5%) compared with normal levels of vitamin D (0%). They supposed that the immune-modulating mechanism of vitamin D or the increasing autophagy might explain the lack or the loss of response to anti-TNF α. [27] Other studies on IBD consistent with these findings led the authors to hypothesize a negative effect of insufficient levels of vitamin D on the retention rate of anti-TNF drugs [35], as well as to highlight the association between low vitamin D levels before anti-TNF therapy and reduction of achievement of remission after three months of treatment [36] or to presume that levels of vitamin D could be associated with disease activity and that the supplementation of vitamin D could increase the serum concentration of infliximab [19,20,21,22,23,24,25]. In contrast, a Canadian prospective interventional study on IBD demonstrated that patients with insufficient levels of vitamin D are more likely to achieve remission during infliximab treatment, supposing that the highest proliferation of T cells makes available more transmembrane TNF-α receptors, improving the b-DMARDs effects of infliximab, increasing apoptosis and consequently decreasing inflammation. [37] Regardless, our study for the first time provided data on the earlier discontinuation of b-DMARDs (in particular anti-TNF-α) in PsA patients with 25 (OH) D ≤ 20 ng/mL, a probable role of vitamin D in improving immune-modulation and immune response, and on contrasting the pro-inflammatory cytokines axes enhancing the mechanism of action of b-DMARDs, which should be taken into account.

As regards methotrexate (MTX) monotherapy in PsA, we evidenced an earlier discontinuation in the group of PsA patients with 25 (OH) D ≤ 20 ng/mL, with a higher risk of discontinuation for failure. MTX’s real effectiveness in PsA is still debated, in particular on enthesitis and axial involvement [38,39]. Of note in our study, patients with sacroiliitis were not treated with MTX according to European and American recommendations. There are no published data on MTX survival in PsA patients with different statuses of vitamin D. Few authors have described a higher rate of adverse events (such as mucositis and testis injuries) in the case of insufficient levels of vitamin D [40,41], but we did not observe differences in the occurrence of adverse events in the groups of PsA patients with diverse statuses of vitamin D.

Due to the retrospective design of this study, some limitations have to be considered, such as selection bias of the patients, treatment management, and incomplete data on the efficacy and severity of PsA. However, the consideration of the presence of sacroiliitis, the drug survival, and the drug discontinuation rate could be contemplated as reliable indexes for estimating PsA severity in real-life practice. It is also important to consider among the limitations that the small sample size of our PsA cohort might limit the statistical power for several analyses. Lastly, the highest prevalence of anti TNF α, chosen as the first b-DMARDs, is basically a consequence of the study period starting in 2010.

## 4. Materials and Methods

We realized a retrospective analysis, evaluating the clinical records of a longitudinal cohort of 570 patients, regularly admitted to our clinic, and then evaluated for follow-up at our outpatient clinic, from 2010 to 2022. The inclusion criteria were the satisfaction of the CASPAR classification criteria for PsA [42] and the presence of evaluation of vitamin D serum levels at baseline visit and at clinical follow-up visits (almost two determinations per year were required). We considered just the patients that maintained their status of vitamin D during the period of the study, despite the doctor’s advice for vitamin D supplementation. The exclusion criteria were age less than 18 years old and the satisfaction of rheumatoid arthritis classification criteria [43] (during the study time). Patients with HLA B27 were also excluded to make the patients more homogeneous in clinical characteristics. We recorded baseline data of patients at the PsA diagnosis time and at follow-up.

Clinical domains, laboratory data (erythrocytes sedimentation rate (ESR)), C-reactive protein (CRP), imaging data (joints X-ray and magnetic resonance imaging), type of joint involvement (oligoarticular or polyarticular), presence of enthesitis, axial involvement (defined by the presence of inflammatory back pain and by radiographic evidence of sacroiliitis or spondylitis), and the disease activity scores (DAS 28 [44] and DAPSA [45]) were collected at baseline visit. The comorbidities were checked at the baseline visit and were reported as Charlson Comorbidity Index [46]. The therapies of patients were listed at baseline and follow-up visits. Serological tests were executed at the PsA diagnosis time according to local guidelines and the cut-off values declared by commercially available assays. In addition, 25(OH)D levels were performed at each patient’s reference laboratory by radio-immunoassays, based on local guidelines, and were always repeated at the same laboratory during follow-up.

To evaluate if the different statuses of serum 25(OH)D levels could interfere with PsA course and treatment response, we defined three study groups of patients, based on other previous studies [23,47,48,49,50]: patients with serum levels of 25(OH)D ≤ 20 ng/mL, patients with serum levels of 25(OH)D between 20 and 30 ng/mL, and patients with serum levels of 25(OH)D ≥ 30 ng/mL.

This study was approved by the local ethics committee (Ethics Review Board of Policlinico of Foggia, protocol number 49/CE/2019), and all patients were informed about the nature and aim of the study and gave and signed their consent to participate in this study.

### Statistical Analysis

The results are shown as mean *±* S.D. (standard deviation) and as a percentage. The normal distribution was evaluated using the Shapiro-Wilk’s test. Comparisons between study groups of PsA patients were assessed by the Student’s *t*-test or by Mann–Whitney U test as opportune. The differences between categorical variables were examined by Pearson chi-square or Fisher’s exact test, as appropriate. The estimation of drug survival was performed by Kaplan–Maier’s estimate, followed by the long-rank (Mentel–Cox) test in the case of comparison between various groups of patients. The Cox regression model was used to evaluate the drug discontinuation risk, which is shown as the hazard ratio (HR) and 95% confidence interval (CI). Statistical significance was fixed at *p* ≤ 0.05. All statistical analyses were realized using IBM SPSS Statistics 26.

## 5. Conclusions

This study definitely confirms the association between vitamin D deficiency and more severe PsA course in terms of the presence of sacroiliitis and MTX poor response in peripheric involvement. However, it newly provides that the 25 (OH) D levels ≤ 20 ng/mL frame a cluster of PsA patients characterized by the lowest retention rate of first b-DMARDs. Further prospective studies are needed to validate these data and, above all, to assess if the supplementation of vitamin D could improve the b-DMARDs response in PsA patients.

## Figures and Tables

**Figure 1 ijms-24-05368-f001:**
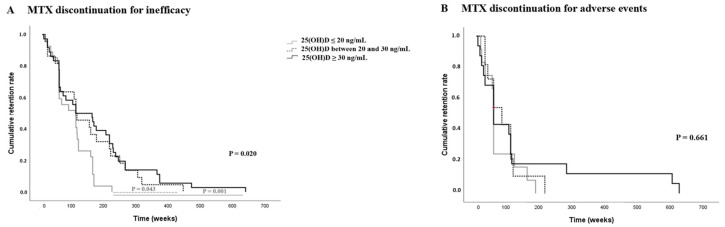
(**A**) and (**B**) Kaplan-Maier method. Methotrexate monotherapy survival in the groups of PsA patients with different serum level statuses of vitamin D: 25(OH)D ≤ 20 ng/mL, 25(OH)D between 20 and 30 ng/mL, and 25(OH)D ≥ 30 ng/mL. (**A**) Discontinuation of methotrexate for inefficacy. (**B**) Discontinuation of methotrexate for adverse events.

**Figure 2 ijms-24-05368-f002:**
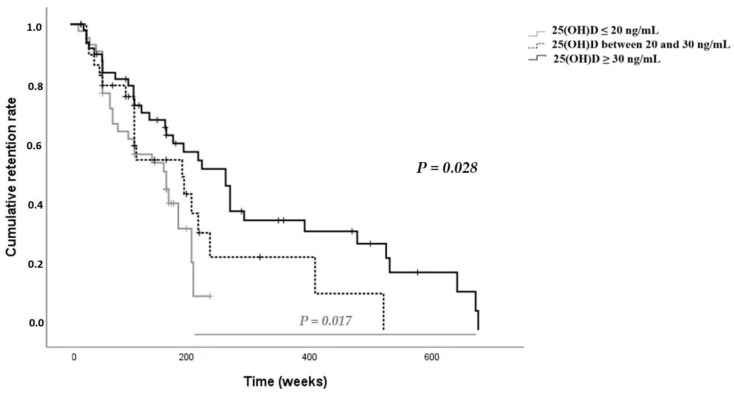
Kaplan–Maier method. First biological drug survival in the groups of PsA patients with different statuses of vitamin D serum level: 25(OH)D ≤ 20 ng/mL, 25(OH)D between 20 and 30 ng/mL, and 25(OH)D ≥ 30 ng/mL.

**Table 1 ijms-24-05368-t001:** Comparison of demographic and clinical features at the diagnosis of PsA time between patients with different statuses of 25(OH)D serum level: 25(OH)D ≤ 20 ng/mL vs. 25(OH)D between 20 and 30 ng/mL vs. 25(OH)D ≥ 30 ng/mL.

	25(OH)D ≤ 20 ng/mL	25(OH)D between 20 and 30 ng/mL	25(OH)D ≥ 30 ng/mL	*p* Value
	N = 90	N = 58	N = 85	
Sex f/m	50 (55.5%)/40 (45.5%)	42 (72%)/16 (28%)	62 (73%)/23 (27%)	0.026
Age at the diagnosis (m ± sd)	57.6 ± 14.4	54.9 ± 12.5	58.8 ± 12.5	0.226
BMI (m ± sd) [range]	27.7 ± 5.6 [25.4–29.9]	27.1 ± 7.2 [24.4–29.9]	26.5 ± 5 [24.9–28.1]	0.723
Smokers n (%)	29 (32%)	18 (31%)	25 (29%)	0.387
Symptoms duration at the diagnosis (years)	4.2 ± 4.8	4.9 ± 5.3	4.3 ± 5	0.735
Oligoarticular involvement n (%)	18 (20%)	14 (24%)	15 (18%)	0.636
Polyarticular involvement n (%)	72 (80%)	44 (76%)	70 (82%)	0.641
Enthesytis n (%)	58 (64%)	39 (67%)	59 (69%)	0.882
Cutaneous psoriasis n (%)	42 (47%)	23 (40%)	48 (56%)	0.129
Sacroiliitis n (%)	31 (34%) °	10 (17%)	7 (8%)	0.0001
DAS 28 (m ± sd)	3.1 ± 1	3.3 ± 0.9	3.2 ± 1	0.771
DAPSA (m ± sd)	21.7 ± 12.5	20.3 ± 12.3	21.6 ± 13.3	0.886
LEI (m ± sd)	2.6 ± 1.4	2.1 ± 1.6	2.3 ± 1.9	0.764
ESR (mm/h)	9.2 ± 15.4	12 ± 16.4	10.3 ± 11	0.768
CRP (mg/l)	4.5 ± 1.3	5 ± 1.6	4.5 ± 1.3	0.365
HAQ (m ± sd)	0.45 ± 0.37	0.43 ± 0.32	0.42 ± 0.30	0.756
Start MTX monotherapy n (%)	68 (75%)	48 (83%)	73 (86%)	0.678
Charlson Comorbidity Index n (%)	1.8 ± 1.5	1.7 ± 1.4	1.8 ± 1.3	0.833
Autoimmune thyroiditis n (%)	6 (7%)	5 (9%)	9 (11%)	0.652
IBD n (%)	5 (6%)	4 (7%)	4 (5%)	0.855
Cardiovascular diseases n (%)	31 (34%)	11 (19%)	21 (25%)	0.098
Diabetes n (%)	13 (14%)	4 (7%)	9 (11%)	0.355
Osteoporosis n (%)	13 (14%)	12 (21%)	7 (8%)	0.251
Osteopenia n (%)	16 (18%)	11 (19%)	15 (17%)	0.414
Total Cholesterol mg/dl (m ± sd)	198.8 ± 40.6	217.5 ± 44.6 *	192.8 ± 33.9	0.007
HDL mg/dl (m ± sd)	51.2 ± 13.6 ^§^	57.6 ± 14.6 ^#^	55.6 ± 15.2	0.046
LDL mg/dl (m ± sd)	126.7 ± 29.4	141.5 ± 39.4	117.2 ± 27.5	0.006
Triglycerides mg/dl (m ± sd)	125.2 ± 50.6	115.7 ± 49.6	133 ± 88.9	0.444
Total cholesterol/HDL ratio (m ± sd)	4 ± 0.9 ^+^	3.8 ± 0.9	3.6 ± 0.9	0.05
Total cholesterol/HDL ratio in female	3.6 ± 0.7	3.6 ± 0.9	3.5 ± 0.9	0.753
Total cholesterol/HDL ratio in male	4.2 ± 1 ^++^	4.1 ± 0.9	3.8 ± 0.8	0.05
Metabolic syndrome n (%)	12 (13%)	16 (10%)	12 (14%)	0.402

Data are shown as mean *±* sd or n (%). BMI: body mass index; CRP, C-reactive protein; ESR: erythrocytes sedimentation rate; HDL: high-density lipoprotein; HAQ: Health Assessment Questionnaire; IBD: inflammatory bowel diseases; LDL: low-density lipoprotein; LEI: Leeds Enthesitis Index; MTX: methotrexate. ° 25(OH)D ≤ 20 ng/mL vs. 25(OH)D ≥ 30 ng/mL: *p =* 0.0001. * 25(OH)D between 20 and 30 ng/mL vs. 25(OH)D ≤ 20 ng/mL: *p =* 0.015; 25(OH)D between 20 and 30 ng/mL vs. 25(OH)D ≥ 30 ng/mL: *p =* 0.002 § 25(OH)D ≤ 20 ng/mL vs. 25(OH)D ≥ 30 ng/mL: *p =* 0.018. # 25(OH)D between 20 and 30 ng/mL vs. 25(OH)D ≤ 20 ng/mL: *p =* 0.049; 25(OH)D between 20 and 30 ng/mL vs. 25(OH)D ≥ 30 ng/mL: *p =* 0.001. + 25(OH)D ≤ 20 ng/mL vs. 25(OH)D ≥ 30 ng/mL: *p =* 0.040. ++ 25(OH)D ≤ 20 ng/mL vs. 25(OH)D ≥ 30 ng/mL: *p =* 0.047.

**Table 2 ijms-24-05368-t002:** Comparison of therapies at follow-up visits between PsA patients with different statuses of 25(OH)D serum level: 25(OH)D ≤ 20 ng/mL vs. 25(OH)D between 20 and 30 ng/mL vs. 25(OH)D ≥ 30 ng/mL.

		25(OH)D ≤ 20 ng/mL	25(OH)D between 20 and 30 ng/mL	25(OH)D ≥ 30 ng/mL	*p* Value
MTX failure		35 (51%)	22 (45%)	37 (50%)	0.340
MTX adverse event		12 (18%)	11 (23%)	14 (19%)	0.485
MTX induced remission		0 (0%)	0 (0%)	3 (3%)	0.198
MTX treatment duration (weeks)		119.8 ± 9.1	163.4 ± 21.1	239.2 ± 27.7	0.098
Start b-DMARDs		47 (52%)	34 (59%)	53 (62%)	0.960
DAS 28 at the start of b-DMARDs		3.9 ± 0.8	3.8 ± 0.7	3.9 ± 0.6	0.235
DAPSA at start of b-DMARDs		25.7 ± 9.6	25.3 ± 7.8	24.5 ± 11.3	0.467
First b-DMARDs failure	Anti-TNF α	18 (100%)	16 (94%)	31 (97%)	0.642
	Anti-IL17	0 (0%)	1 (6%)	1 (3%)	0.533

Data are shown as mean *±* sd or n (%). b-DMARDs, biological drug; DAPSA: Disease Activity Index for Psoriatic Arthritis, DAS: disease activity score. MTX, methotrexate.

## Data Availability

The data presented in this study are available on request from the corresponding author.

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
