# Peer review of "Vitamin D Status and Psoriatic Arthritis: Association with the Risk for Sacroiliitis and Influence on the Retention Rate of Methotrexate Monotherapy and First Biological Drug Survival—A Retrospective Study"

_ijms, 2023, doi:10.3390/ijms24065368_

Round 1

Reviewer 1 Report (Previous Reviewer 1)

The manuscript improved by the revision of the authors.

Two minor suggestions, I would advise to stick to b-DMARDs (biological disease modifying anti-rheumatic drugs) throughout the manuscript as usual in the literature. 

Abstract:

"The important role of vitamin D as a determinant factor in the clinical presentation, in particular sacroiliac involvement, and on the drug (methotrexate and biological therapies) survivals in PsA patients is clearly proved in our study."

 due to the study sign, there no clear proof is possible. Therefore I would suggest du change it to something like that:

This study highlights significant differences in clinical presentation, in particular sacroiliac involvement and on drug survival (methotrexate and b-DMARDs) in PsA patients with vitamin D deficiency.

Or something like that.

Author Response

  • Two minor suggestions, I would advise to stick to b-DMARDs (biological disease modifying anti-rheumatic drugs) throughout the manuscript as usual in the literature. 
  • Answer: we change as requested

  • Abstract: "The important role of vitamin D as a determinant factor in the clinical presentation, in particular sacroiliac involvement, and on the drug (methotrexate and biological therapies) survivals in PsA patients is clearly proved in our study." due to the study sign, there no clear proof is possible.

Therefore I would suggest du change it to something like that: This study highlights significant differences in clinical presentation, in particular sacroiliac involvement and on drug survival (methotrexate and b-DMARDs) in PsA patients with vitamin D deficiency.

Or something like that.

  • Answer: we change as requested

Reviewer 2 Report (New Reviewer)

Introduction. Comments on fracture risk and vitamin D are not relevant to this paper and could be deleted. Paragraph 3 is rather unclear and should be revised. 

The authors have analysed a cohort of 570 patients with PsA to look at the effects of vitamin D levels. Only 233 patients were used but it is not clear what the main reasons for exclusion were, though several criteria are listed. This should be clarified. 

The main difficulty I see in the method of analysis is the division of the patients into three groups. This seems arbitrary and may not be statistically valid. There are no details on what statistical methods were used in the analysis given in methods, and whether a statistician was consulted (I write as a non-expert statistician). However, for some analyses I think it would be better to use the whole dataset and look for correlations between vitamin D levels as a continuous variable and other measures of interest e.g. CRP, ESR etc, although few differences between groups were reported anyway. For sacroiliitis it should be possible to look at correlation with degree of change on radiographs and vitamin D, rather than just present/absent. The trends for more comorbidities in the lower vitamin D group are not impressive and should be omitted.

The main other finding is the effect of vitamin D level on duration of treatment with MTX and then a first biological drug. The initial analysis of MTX is confusing and I can't see the p values quoted in the text in the corresponding tables, but only in the Kaplan-Meier graph where the differences seem very small despite the p value. Again analysis of vitamin D levels as a continuous variable vs survival in days would be better. The differences in the duration of anti-TNF therapy are more impressive but again additional statistical support for the conclusions would be helpful.

Author Response

  • Comments on fracture risk and vitamin D are not relevant to this paper and could be deleted.
  • Answer: we delete the considerations on fracture risk and vitamin D.
  •  
  • Paragraph 3 is rather unclear and should be revised. 
  • Answer: we revised paragraph 3
  •  
  • The authors have analysed a cohort of 570 patients with PsA to look at the effects of vitamin D levels. Only 233 patients were used but it is not clear what the main reasons for exclusion were, though several criteria are listed. This should be clarified. 
  • Answer: we explain the cause of exclusion.

The main difficulty I see in the method of analysis is the division of the patients into three groups. This seems arbitrary and may not be statistically valid. There are no details on what statistical methods were used in the analysis given in methods, and whether a statistician was consulted (I write as a non-expert statistician). However, for some analyses I think it would be better to use the whole dataset and look for correlations between vitamin D levels as a continuous variable and other measures of interest e.g. CRP, ESR etc, although few differences between groups were reported anyway. For sacroiliitis it should be possible to look at correlation with degree of change on radiographs and vitamin D, rather than just present/absent. The trends for more comorbidities in the lower vitamin D group are not impressive and should be omitted. The main other finding is the effect of vitamin D level on duration of treatment with MTX and then a first biological drug. The initial analysis of MTX is confusing and I can't see the p values quoted in the text in the corresponding tables, but only in the Kaplan-Meier graph where the differences seem very small despite the p value. Again analysis of vitamin D levels as a continuous variable vs survival in days would be better. The differences in the duration of anti-TNF therapy are more impressive but again additional statistical support for the conclusions would be helpful.

  • Answer: we add the “statistical analysis” section. The analysis was performed by an expert statician Of note, the division into the three groups reflects previous study and the values of vitamin D noted for deficiency and hypovitaminosis D. We add the linear correlation between vitamin D and other measures of interest (CRP, ESR, MTX and b-DMARDs treatment duration). Of note, the Acute phase reactants in psoriatic arthritis have less significance on disease status compared to rheumatoid arthritis. We add the MTX treatment duration in the table.

Reviewer 3 Report (New Reviewer)

General comments:

The authors reported that a clinical difference between a serum values of Vit D and PsA clinical course.

As they stated, the mode of action of Vit D to response of PsA was very complexed and it was unclear how Vit D interacted with pathophysiological phase in PsA. 

I agree with a special importance of Vit D in human, however, it is hard to report to connect Vit D and pathological and clinical events in PsA.

The authors should clarify more clinicopharmacological elucidations in mechanisms of Vit D on sacroiliitis in PsA. 

Specific comments:

1. Was metabolism of Vit D correlated with clinical phenomena?

2. Vit D is fat soluble. Was adipose tissue thought with Vit D?

3. Was discontinuation of MTX or biologic due to disease activity?

4. How did you rule out effects of other metabolites out of Vit D?

5. Multiple factors to analyze the association with the risk for sarcoiliitis and the influence of medications should not have been.

Author Response

General comments:

  • The authors reported that a clinical difference between a serum values of Vit D and PsA clinical course. As they stated, the mode of action of Vit D to response of PsA was very complexed and it was unclear how Vit D interacted with pathophysiological phase in PsA. I agree with a special importance of Vit D in human, however, it is hard to report to connect Vit D and pathological and clinical events in PsA. The authors should clarify more clinicopharmacological elucidations in mechanisms of Vit D on sacroiliitis in PsA. 
  • Answer: we add some considerations in the discussion section.

Specific comments:

  1. Was metabolism of Vit D correlated with clinical phenomena?
  2. Vit D is fat soluble. Was adipose tissue thought with Vit D?

Answer: the BMI did not differ in different groups

  1. Was discontinuation of MTX or biologic due to disease activity?

Answer: we described the different reasons of discontinuation in the results (inefficacy and adverse events)

  1. How did you rule out effects of other metabolites out of Vit D?

Answer: None of our patients presented kidney insufficiency or kidney disease, so the evaluation of 25 (OH)D level is sufficient.

  1. Multiple factors to analyze the association with the risk for sarcoiliitis and the influence of medications should not have been.

Answer: As regard the influence on risk for sacroiliitis we exclude HLA B27+ patients that is the principal risk factor for axial and sacro-iliiitis risk in patients. As regard the influences on medication we consider the discontinuation for adverse event and inefficacy. In addiction we operated a gender analysis that exclude possible influoces of gender.

Round 2

Reviewer 2 Report (New Reviewer)

The authors have described the statistical techniques used and completed additional analysis which supports their conclusions.

Reviewer 3 Report (New Reviewer)

No comments

This manuscript is a resubmission of an earlier submission. The following is a list of the peer review reports and author responses from that submission.

Round 1

Reviewer 1 Report

In this study by Rotondo et al. a retrospective study was performed and Vitamin D status potentially identified as a risk factor for clinical presentation of PsA and the influence on drug survival.

The study has many limitations, far more than stated in the limitations section.

+ the authors state, that patients had no signs for osteomalacia - how? no information on alkaline phosphatase or parathyroid hormone levels were presented.

+ no information on smoking status of patients is presented. Since there is a known association of smoking and low vitamin d levels and also with higher activity of rheumatologic disease activity, this is a very important fact.

+ biotechnological drugs is not appropriate since only biological DMARDs have been investigated, this would be the correct phrasing.

+ in IBD patients low levels of vitamin D are much more common compared to PsA patients alone. Level correlate not only with bowel resection but also with disease activity, therefore this could be a potential bias, however the amount of iBD patients was more or less the same between the 3 groups.

+ MTX as a firstline drug in PsA is not very effective, especially since the amount of patients with sacroiliitis was very high, especially in the first group. So, this is a major confounding factor for earlier treatment cessation. Further, patients with polyarticular arthritis and enthesitis are not effectively treated by MTX, therefore the high amount of MTX stoping is not surprising in this population. The question for me is "hen or egg" since those patients are more severely ill and therefore with a known lower vitamin D level.

+ the authors state in the discussion, that "Anyway, our study for the first time provided data on earlier discontinuation of b-DMARDs (both anti TNF-α and anti IL-17) in 259 PsA patients with 25 (OH) D ≤ 20 ng/ml" --> this is not clear. In total 233 patients were investigated, 47 with 25 (OH) D ≤ 20 ng/ml started bDMARDs, 0/none of the IL-17i group failed firstline. 

+ if there is the hypothesis that systemic inflammation is associated with vitamin D level (and I don't say it isn't, but I am afraid this study is not providing sufficient information to conclude that), how do the authors conclude that there is no difference in cutaneous psoriasis e.g. or do have patients nail psoriasis, further we do not know anything about subclinical inflammation of entheses...

+ and further, Vitamin D level vary throughout the year even if patients are chronic insufficient - no information on time point of evaluation is provided.

Reviewer 2 Report

The manuscript titled: Vitamin D status and psoriatic arthritis: association with clinical domains, the risk for sacroiliitis and influence on the retention rate of methotrexate monotherapy and first biotechnological drug survival. A retrospective study (ijms-2150222) analyzes different statuses of vitamin D in methotrexate monotherapy discontinuation and biotechnological drug survival in PsA.

The manuscript is well-written and presents interesting aspects of PsA; however, a few things should be explained:

1.       What were ECR and CRP in this study? Did these parameters correlate with vitamin D levels? Have you seen differences between ESR and CRP in different statuses of vitamin D?

2.       The authors found a higher total cholesterol/HDL ratio in patients with 25 (OH)D ≤ 20 ng/mL serum level compared to those with 25(OH)D ≥ 30 ng/mL serum level (p=0.040); however, such differences can be related to the BMI value, which was the lowest in a group with vitamin D over 25(OH)D ≥ 30 ng/mL. HDL levels are different for females and males; thus, have you seen differences in cholesterol fractions between women and men?

3.       The age of the analyzed patients is about 57 years. How many women and men are in specific groups? After menopause, the risk of osteoporosis is higher and is related to vitamin D levels. Thus, the analysis according to gender should be done and commented on in this manuscript.  

If the authors consider the above comments, I believe the work is valuable and can be published in the International Journal of IJMS. 

Reviewer 3 Report

The present article deals with an interesting and yet new topic, namely the correlations between the level of vitamin D and various aspects of the evolution of patients with psoriatic arthritis.

Regarding the methodology,

- what is the reason why patients with HLA B27 Ag positivity were excluded even though they fulfilled the CASPAR criteria (with sensitivity and specificity of >95%)

- what exactly does this means: We considered just the patients that maintained their 84 status of vitamin D during the period of the study, despite doctor’s advice for vitamin D 85 supplementation”

- what was the time gap between the baseline and the other evaluations?

- were the patients who took active metabolites of vitamin D during those who did not require renal activation also excluded? (their administration does not change the level of 25(OH) vitamin D)

The title also mentions correlations with different domains of psoriatic arthritis, however no correlation with some of these (enthesitis, dactylitis, skin and nail damage) has been made